# Vaccination coverage estimation in Mexico in children under five years old: Trends and associated factors

**Maria Jesus Rios-Blancas[1], Hector Lamadrid-Figueroa[1]\*, Miguel Betancourt-Cravioto[2], Rafael Lozano[3]**

**1** National Institute of Public Health, Cuernavaca, Morelos, Mexico, **2** Carlos Slim Institute, Mexico City, Mexico, **3** Institute for Health Metrics and Evaluation, University of Washington, Seattle, WA, United States of America

\* hlamadrid@insp.mx

**Data Availability Statement:** All surveys files are available from the INSP database: https://ensanut.insp.mx/encuestas/ensanut2018/descargas.php.

## Abstract

We aimed to estimate vaccination coverage and factors associated in completing schemes in children under 5 years old between 2000 and 2018. A secondary analysis was carried out on five national health surveys between 2000 and 2018 in Mexico. The sample was 53,898 children under 5 years old, where 30% of missing vaccination information was imputed using chained equations. During this period two basic vaccination schemes (CBS) were identified. For each doses and vaccines of both schemes and completed CBS, the coverage was estimated using weighted logistic regression models. Additionally, the factors associated with incomplete schemes were reported. Between 2000 and 2018, the caretakers who did not show the vaccination card went from 13.8% to 45.6%. During this period, the estimated vaccination coverages did not exceed 95%, except for BCG and marginally the first doses of vaccines against pneumococcus, acellular pentavalent, and Sabin. In the same period, the CBS estimated coverage decreased steadily and was under 90%, except for children aged 6–11 months (92.6%; 91.5–93.7) in 2000. Not having health insurance stands out as an associated factor with incomplete vaccination schemes. In conclusion, the imputation allowed to recuperate information and obtain better data of vaccination coverage. The estimated vaccination coverage and CBS do not reach sufficient levels to guarantee herd immunity, hence innovative strategies to improve vaccination must be established in Mexico.

## Introduction

Vaccination is one of the most cost-effective public health interventions, it has significantly reduced morbidity and mortality, particularly in children under 5 years old [1]. According to the estimates from the Global Burden Disease study, in this age group, globally and in the Latin American (LA) region, deaths from vaccine-preventable diseases (VPD) went from 5.1 to 1.8 million and from 228 to 36 thousand, respectively between 1990 and 2017. In the same period, the decrease in years of healthy life by VPD was 67.8% (482.2 to 155.2 million) worldwide and 83.0% (20.3 to 3.4 million) in LA [2].

**Funding:** The authors received no specific funding for this work.

**Competing interests:** The authors have declared that no competing interests exist.

To consolidate strategies to reduce the burden generated by VPD, various organizations and institutions around the world have implemented programs and aligned their agendas. According to the Expanded Program on Immunization (EPI), which was established by the World Health Organization (WHO) in 1974, more than 1 billion children were vaccinated over the last decade, allowing the avoidance of 2–3 million deaths every year. The Americas, after the adoption of EPI by the Pan American Health Organization (PAHO), became the first of the WHO regions to eliminate poliomyelitis in 1994, then was declared free of measles, rubella, and congenital rubella syndrome (2015–2016) and neonatal tetanus (2017), among others achievements [3]. Mexico was not the exception; one year before the creation of EPI, it established the National Immunization Program where it organized mass vaccination and simultaneous application of five vaccines against six diseases [4]. This scheme has evolved. Nowadays it covers the population against fourteen diseases, thus distinguishing itself as one of the most complete in LA region [4].

The vaccination coverage–percentage of vaccinated children in an age group in regard to the population in that age group–is an incomplete but helpful indicator in evaluating vaccination programs [5–7]. It is incomplete because ideally the vaccination should be monitored with immunity information; nevertheless, due to its high cost there is scarce and/or untimely data [8, 9]. However, vaccination coverage is a widely used indicator, which supports decisions-making about the progress and challenges for the fulfillment of global, regional, or national goals, such as polio eradication and measles elimination [10, 11]. This indicator can help in the allocation of financial sources, distribution of human resources, and introduction of new vaccines, among others [9, 12].

The main information sources for monitoring vaccination coverage are registries or administrative reports [9], electronic immunization registries [13] and/or surveys [5, 7]. "Administrative coverage" as calculated in most low-and middle-income countries, gives information about the number of doses administered in the target population, commonly derived from census estimates [9]. Nevertheless, it tends to overestimate vaccinations due to duplicate records, errors in transcripts, inaccurate population under responsibility, etc. Electronic records, although providing continuous information of coverage, vaccine supply, vaccination reminders, etc.; [13] present challenges in their implementation and sustainability, particularly in low- and middle-income countries [14, 15].

Surveys represent a widely used resource to validate the administrative information systems coverage results, where an important data source is the vaccination card (VC), also called health card or home-based record [5, 16]. Recently, EPI group experts published a strategic report for the collection, processing, analysis, and vaccination coverage report according to household surveys. This report is focused on routine vaccination activities and highlights the VC as the main source of information and in its absence, the reminder response was by the children's mother or guardian. They present some recommendations to calculate coverage: first, give the same power to both sources and add them; second, add to the denominator those children without vaccine information and do not count them in the numerator; finally, use the imputation methods to estimate vaccination status in the group without a VC [17].

There is evidence that vaccination coverage based on administrative reports has several disadvantages [5, 18], such as in Mexico, where this information is not reliable for estimating vaccination due to constant overestimation [19] and has presented significant differences with the survey results [20]. On the other hand, although the survey continues to provide important information in the short-to medium term, it has also disadvantages [5, 7, 17]. In Mexico, where the vaccination coverage measurement according to the surveys is based on those children who had VC and whose mothers or guardians showed or proved it [20–22], has been steadily decreasing; thus, inferences and decisions protecting the population against VPD and the program operation may be affected.

In this regard, this study aimed to estimate Mexican vaccination coverage in children under five years old between 2000 and 2018, adjusting for the biases introduced by the lack of information on children who did not have or did not show the vaccination card.

## Materials and methods

### Data sources

A secondary analysis from five Mexican national health surveys (2000 [23], 2006 [24], 2010 [25], 2012 [26], and 2018 [27]) was carried out, which design was probabilistic with national and state representativeness. The surveys content and protocols were approved by the ethics and research committee of the National Institute of Public Health (INSP, by its acronym in Spanish) of Mexico. Both INSP and the National Institute of Statistic and Geography (INEGI, by its acronym in Spanish) carried out data collection; household questionnaires and individual questionnaires were used [23–27]. All participants in the surveys provided oral and/or written consents, where information was provided on the objectives of the research, the voluntary nature of participation and the confidentiality of the information. For children under five years old, the mother or guardian (caretaker) was asked to provide information on the vaccination of the children. All the information collected was anonymized and published in digital repositories. More detail was published elsewhere [23–27].

For the analysis, 53,898 records from children under 5 years old whose caretaker provided information on the child's vaccination through the vaccination card or by recall were reviewed. In addition, sociodemographic information on the child, the caretaker, as well as the characteristics of the household of residence were considered.

### Variables

The analyzed variables provide information about the dosage, vaccine application and adherence to the complete basic scheme (CBS). During the study period, two vaccination schemes were identified excluding booster doses [4]. Summarizing, there were 22 dependent variables (DV), which corresponded to the eight doses from the first scheme, 12 from the second, and two dose-indicators (one for each CBS).

The first scheme is valid during the 2000 and 2006 surveys, including Bacillus Calmette and Guerin (BCG), pentavalent, Sabin and vaccine against measles, mumps, and rubella (MMR). The second scheme, valid for the 2010, 2012 and 2018 surveys, includes BCG, pentavalent acellular, against hepatitis B, pneumococcus, rotavirus, and MMR [4]. Table 1 shows the recommended application age for each dose. Considering that most of these doses are applied during the first 11 months of life, four age groups were defined for children under one-year-old and other groups were defined for children of one, two, three, and four years old. Therefore, the corresponding doses estimated coverage is presented for each of these age groups.

The variables indicating adherence to the basic scheme, meaning they have the indicated doses or missing at least one of the doses. They are reported at the state level and for two age groups: 6 to 11 months, where seven out of eight doses (first scheme), and 11 out of 12 doses (second scheme) must be applied; and 12 to 23 months, where all children should have received all their doses.

### Covariates

The covariates from the child that were considered for the analysis were sex and age in months. For the caretaker they were age (under 20 years old; 20 and older), sex, schooling (elementary or none; middle school or higher), health insurance (HI) (with HI [formal workers sector],

**Table 1. Accumulated doses in the two complete basic schemes valid between 2000 and 2018 by age group.**

| Age group | Complete Basic Scheme 1 [a] (accumulated doses) | Complete Basic Scheme 2 [b] (accumulated doses) |
|---|---|---|
| Under two months old | BCG (1 dose) | BCG + 1HB (2 doses) |
| 2–3 months | BCG + 1 PV + 1 Sabin (3 doses) | BCG + 2HB (or 1HB if the first one was not given at birth) + 1 PVa + 1 Pnm + 1 RV (5 or 6 doses) |
| 4–5 months | BCG + 2 PV + 2 Sabin (5 doses) | BCG + 2HB + 2 PVa + 2 Pnm + 2 RV (9 doses) |
| 6–11 months | BCG + 3 PV + 3 Sabin (7 doses) | BCG + 3HB + 3 PVa + 2 Pnm + 2 RV (11 doses) |
| 12–23 months | BCG + 3 PV + 3 Sabin + MMR (8 doses) | BCG + 3HB + 3 PVa + 2 Pnm + 2 RV + MMR (12 doses) |
| 24–35 months | | |
| 36–47 months | | |
| 48–59 months | | |

BCG, Bacillus Calmette-Guerin vaccine; PV, complete cells pentavalent vaccine; PVa, acellular pentavalent vaccine; HB, Vaccine against Hepatitis B; Pnm, Conjugate vaccine against Pneumococcal; RV, Vaccine against rotavirus; MMR, Vaccine against measles, mumps, and rubella.

[a] Vaccination scheme valid during the 2000 and 2006 surveys.

[b] Vaccination scheme valid during the 2010, 2012 and 2018. The boost doses were not included (3 Pnm, 3 RV y 4 Pva).

without HI [private sector, Secretary of Health]) and indigenous status. Finally, for the household of residence: rurality, and state (which was categorized into four geographic regions: northwest, northeast, center, and south).

## Imputation procedure

Multiple imputation (MI) was proposed by Rubin (1986) as a method to address the missing information or data absence. Briefly, the strategy consists of generating a different value number for each missing data to maintain the population variability and maintain the relationship between variables. The theoretical foundation of MI is based on Bayesian methods [28, 29].

A multiple imputation process by chained equations (MICE) was carried out for this study, where it was considered that the information loss pattern was not completely random [28–30]. Children whose caretakers did not present the VC at the survey time were defined as missing data in the DV, as well as the covariates were those without values.

Imputation model (1) was adjusted by the independent variables described above and survey design. Then, internal validation tests were performed using Bootstrap techniques [31, 32] and two variants of the number of database replications (m) [33].

$$f(Q|Y_{obs}) = \int f(Q|Y_{obs}, Y_{mis}) f(Y_{mis}|Y_{obs}) dY_{mis} \tag{1}$$

Where:

$f(Q|Y_{obs})$: Final distribution of parameter Q given the observed data.

Q: Proportion.

$f(Q|Y_{obs}, Y_{mis})$: Distribution of parameter Q given the complete data.

$f(Y_{mis}|Y_{obs})$: Distribution of missing data given observed data.

$\int dY_{mis}$: Integral regarding the distribution of missing data.

Then, the estimator and associated variance (T) were obtained from the model with the best performance and considering the following Rubin rules:

$$\bar{Q} = \frac{1}{m} \sum_{j=1}^{m} \hat{Q}_j \tag{2}$$

$$T = \bar{U} + B + \frac{B}{m} \qquad (3)$$

Where this total variance (T) constitutes the variability within ($\bar{U}$) and between (B) the $m$ replications preformed:

$$\bar{U} = \frac{1}{m} \sum_{j=1}^{m} U_j \qquad (4)$$

$$B = \frac{1}{m} \sum_{j=1}^{m} (\hat{Q}_j - \bar{Q})^2 \qquad (5)$$

## Analysis

In order to evaluate the groups' homogeneity with and without VC in the first stage, a general description from the variable of interest in the surveys was carried out and they were compared using the Pearson's Chi$^2$ test, adjusting for the surveys´ design.

The missing information was imputed as described above. The probability of applying the 22 doses and two CBS ($\hat{Y}_i$) was estimated to report the estimated vaccination coverage. Logistic regression models adjusted by the covariates described above and survey year were specified for the vaccination coverage purpose. Additionally, two interaction terms were included: caretaker's sex and age and the survey year with the state. The analysis was adjusted by each year survey design and imputed information. Estimated coverages according to dose and CBS by age group, and the survey year were reported.

Aiming to identify determinants of the incomplete basic vaccination scheme from the two age groups (6–11 months and 12–23 months), the odds ratios were estimated with their respective confidence intervals for the variables: rurality, age, schooling, health insurance, and the caretaker indigenous condition. These estimates are reported globally, meaning the five surveys were included.

The models´ descriptive, imputation, and analysis were carried out with the statistical package STATA version 15 [34], while for the validation tests the R version 3.6.2 was used [35]. The confidence level and significance level were 95% and 0.05, respectively.

## Results

### The surveys´ non-response exploration

The caretaker proportion who showed VC at the survey time decreased from 82.5% to 49.1% in 2000 and 2018, respectively, while the group that reported having VC but did not show it increased from 13.8% to 45.6% for the same years (Fig 1).

Characteristics of the children, caretakers, and household of residence between the populations that showed VC and those that did not show it, by each year survey, were heterogeneous. Overall, the children's average age was higher and statistically significant in the group that did show VC. Regarding the caretakers, a higher proportion of women 20–40 years older, not indigenous and without health insurance were found (Table 2).

### Estimated vaccination coverage by age group

**Under two months old.** The BCG vaccine is present in both schemes and its estimated coverage exceeded 90% in the study period. Furthermore, the trend was maintained in the other age groups. In contrast, the first Hepatitis B (HB) dose vaccine presented a downward

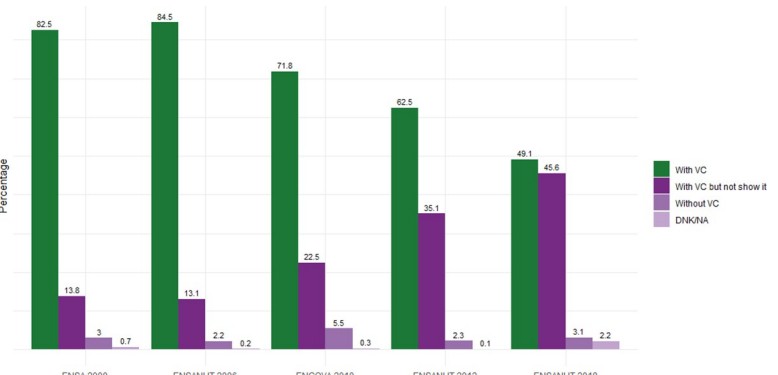

**Fig 1. Information type trend provided by national health surveys, Mexico 2000–2018.** VC, Vaccination Card; DNK/NA, Does not know, No answer. Acronyms in Spanish: ENSA, Encuesta Nacional de Salud (National Health Survey); ENSANUT, Encuesta Nacional de Salud y Nutrición (National Health and Nutrition Survey); ENCOVA, Encuesta Nacional de Cobertura de Vacunación (National Coverage Survey).

trend but it was not significant; it went from 83.2% (95% CI: 77.1–89.3) in 2010 to 82.7% (95% CI 77.7–87.7) in 2018 (Table 3).

**Two to three months old.** The vaccines first dose estimated coverage in scheme one, Sabin, and pentavalent (PV) were greater than 60% but it decreased between 2000 and 2006. Regarding the second scheme, the first dose estimated coverage of acellular pentavalent (PVa), of pneumococcus (Pnm) and rotavirus (RV) was also greater than 60%. The HB first dose was greater than the HB second dose, which decreased significantly from 71.9% (95% CI: 63.8–80.0) in 2010 to 17.6% (95% CI: 13.7–21.5) in 2018 (Table 3).

**Four to five months old.** Globally, it is observed that the second dose vaccine estimated coverage in both schemes was less than the first dose. Furthermore, while these decreased in the study period, the Pnm vaccine showed a significant increase between 2010 and 2018: the first dose went from 82.7% (95% CI: 77.3–88.0) to 96.2% (95% CI: 94.3–98.2) and the second dose from 46.6% (95% CI: 37.3–56.0) to 77.6% (95% CI 70.0–85.3) (Table 3).

**Six to 11 months old.** We observed that the third dose estimated coverage in both schemes was less than the second dose and this one less than the first dose; this pattern is maintained in the six to eleven month age group. Like the previous group, the estimated coverage decreased with the exception of the Pnm vaccine, whose first dose went from 87.6% (95% CI: 83.9–91.2) in 2010 to 97.5% (95% CI: 96.5–98.4) in 2018, and the second dose, from 71.7% (95% CI: 67.1–76.2) in 2010 to 91.8% (95% CI: 88.4–95.1) in 2018 (Table 3).

**One, two, three and four years old.** The first scheme estimated coverage was greater than 90%, apart from MMR which went from 81% (95% CI: 71.5–90.5) in 2000 to 64.9% (95% CI: 52.5–77.4) in 2018 for one-year old group. The main fluctuations and low estimated coverage were observed in the second scheme, particularly in the HB vaccine second and third doses, as well as MMR, which presented significant decreases in the other three age groups (Table 3) (S1 Table).

The highest values of the CBS estimated coverage in the 6 to 11 months old group date from 2000, where 29 of the 32 states were greater than 90%. This estimated coverage decreased steadily in the following years. All states had estimated coverage of less than 32% in 2018 with the lowest being concentrated in the Mexican northern region (Fig 2). A similar pattern and trend were identified for the one-year old group, where seven of the 32 states had estimated coverages greater than 80% in 2000, and only two states were greater than 50% in 2018 (Fig 3).

**Table 2. Socio-demographic characteristics comparison of the caretaker and the child between the groups with and without VC, according to national health surveys.**

| | ENSA 2000 | | | ENSA 2006 | | | ENCOVA 2010 | | | ENSANUT 2012 | | | ENSANUT 2018 | | |
|---|---|---|---|---|---|---|---|---|---|---|---|---|---|---|---|
| | With VC | Without VC | | With VC | Without VC | | With VC | Without VC | | With VC | Without VC | | with VC | without VC | |
| N | 9,395 | 1,993 | | 9,012 | 1,652 | | 6,460 | 2,539 | | 8,180 | 4,886 | | 5,060 | 4,812 | |
| | n (%) | n (%) | P–value [a] | n (%) | n (%) | P—value [a] | n (%) | n (%) | P—value [a] | n (%) | n (%) | P—value [a] | n (%) | n (%) | P—value [a] |
| **Child Characteristics** | | | | | | | | | | | | | | | |
| **Age (months)** | | | | | | | | | | | | | | | |
| Average (SD) | 29.6 (0.3) | 31 (0.6) | 0.02 | 30.9 (0.3) | 31.3 (0.6) | 0.52 | 29.5 (0.3) | 28.8 (0.6) | 0.27 | 29.3 (0.3) | 32.3 (0.4) | <0.001 | 30.1 (0.3) | 32.1 (0.3) | <0.001 |
| **Sex** | | | | | | | | | | | | | | | |
| Female | 4,582 (48.8) | 1,009 (51.8) | 0.10 | 4,362 (48.4) | 831 (52.2) | 0.06 | 3,127 (47.8) | 1,196 (45.5) | 0.19 | 4,070 (48.5) | 2,443 (50.2) | 0.17 | 2,442 (48.7) | 2,310 (48.7) | 1 |
| Male | 4,864 (51.2) | 992 (48.2) | | 4,650 (51.6) | 821 (47.8) | | 3,311 (52.2) | 1,337 (54.5) | | 4,110 (41.1) | 2,443 (49.7) | | 2,618 (51.3) | 2,502 (51.3) | |
| **Caretaker Characteristics** | | | | | | | | | | | | | | | |
| **Sex** | | | | | | | | | | | | | | | |
| Female | 6,302 (68.1) | 1,412 (71.2) | 0.05 | 8,395 (92.8) | 1,445 (88.1) | <0.01 | 3,660 (58.8) | 1,417 (56.1) | 0.11 | 7,660 (93.0) | 4,253 (87.1) | <0.001 | 4,850 (96.4) | 4,374 (91.4) | <0.001 |
| Male | 3,144 (31.9) | 589 (28.8) | | 617 (7.2) | 207 (11.9) | | 2,777 (41.2) | 1,114 (43.9) | | 520 (7.0) | 633 (12.9) | | 210 (3.6) | 438 (8.6) | |
| **Age** | | | | | | | | | | | | | | | |
| Under 20 years old | 145 (1.0) | 36 (1.1) | <0.01 | 679 (5.9) | 93 (6.4) | <0.001 | 671 (10.1) | 253 (9.5) | 0.74 | 647 (7.4) | 295 (5.6) | <0.001 | 332 (6.9) | 247 (5.5) | <0.001 |
| 20–40 years old | 7,430 (80.8) | 1,481 (76.1) | | 6,965 (80.0) | 1,152 (71.3) | | 4,540 (69.8) | 1,755 (70.8) | | 6,566 (80.8) | 3,628 (74.1) | | 4,194 (83.4) | 3,836 (80.1) | |
| + 40 years old | 1,870 (18.2) | 483 (22.8) | | 1,365 (14.1) | 407 (22.4) | | 1,226 (20.1) | 523 (19.8) | | 967 (11.8) | 962 (20.4) | | 534 (9.7) | 729 (14.5) | |
| **Indigenous Population** | | | | | | | | | | | | | | | |
| Yes | 781 (8.0) | 180 (11.9) | <0.01 | 1,914 (22.7) | 295 (19.3) | 0.04 | - | - | - | 2,261 (24.7) | 1,074 (20.0) | <0.001 | 524 (10.9) | 215 (4.4) | <0.001 |
| No | 8,612 (92.0) | 1,812 (88.1) | | 7,077 (77.4) | 1,348 (80.7) | | - | - | | 5,919 (75.3) | 3,812 (80.0) | | 4,536 (89.1) | 4,597 (95.7) | |
| **Schooling** | | | | | | | | | | | | | | | |
| None and elementary | 4,275 (49.0) | 955 (54.6) | <0.01 | 3,769 (45.7) | 675 (42.4) | 0.10 | 2,047 (35.3) | 628 (27.8) | <0.001 | 2,697 (30.6) | 1,506 (28.5) | 0.08 | 945 (20.1) | 765 (16.8) | <0.01 |
| Middle school and higher | 4,968 (51.0) | 947 (45.4) | | 5,222 (54.3) | 971 (57.6) | | 3,974 (64.7) | 1,697 (72.2) | | 5,483 (69.4) | 3,380 (71.5) | | 4,115 (79.9) | 4,047 (83.2) | |
| **Health Insurance** | | | | | | | | | | | | | | | |
| Yes | 5,750 (62.9) | 1,259 (67.0) | 0.01 | 6,137 (71.0) | 1,065 (67.4) | 0.05 | 4,654 (74.0) | 1,739 (72.4) | 0.29 | 5,994 (71.0) | 3,310 (66.3) | <0.01 | 3,599 (73.2) | 2,964 (62.0) | <0.001 |
| No | 4,025 (37.1) | 808 (33.0) | | 2,867 (29.0) | 585 (32.6) | | 1,806 (26.0) | 798 (27.6) | | 2,183 (29.0) | 1,575 (33.7) | | 1,445 (26.8) | 1,839 (38.0) | |
| **Caretaker Household Characteristics** | | | | | | | | | | | | | | | |
| **Stratum** | | | | | | | | | | | | | | | |
| Urban | 6,940 (74.0) | 1,464 (68.3) | <0.01 | 6,474 (71.9) | 1300 (79.8) | <0.001 | 4,196 (68.5) | 1,882 (78.7) | <0.001 | 4,971 (71.0) | 3,435 (78.8) | <0.001 | 3,057 (63.0) | 3,571 (78.9) | <0.001 |
| Rural | 2,506 (26.0) | 537 (31.7) | | 2,538 (28.1) | 352 (20.2) | | 2,242 (31.5) | 651 (21.4) | | 3,209 (29.0) | 1,451 (21.2) | | 2,003 (37.0) | 1,241 (21.2) | |
| **Region** | | | | | | | | | | | | | | | |
| Northwest | 2,283 (19.7) | 531 (20.3) | 0.01 | 2,152 (20.2) | 468 (23.8) | 0.06 | 1,269 (16.4) | 675 (23.9) | <0.001 | 1,856 (18.6) | 1,267 (23.5) | <0.001 | 819 (13.5) | 1,405 (28.4) | <0.001 |

(*Continued*)

**Table 2.** (Continued)

| | ENSA 2000 | | | ENSA 2006 | | | ENCOVA 2010 | | | ENSANUT 2012 | | | ENSANUT 2018 | | |
|---|---|---|---|---|---|---|---|---|---|---|---|---|---|---|---|
| | With VC | Without VC | | With VC | Without VC | | With VC | Without VC | | With VC | Without VC | | with VC | without VC | |
| N | 9,395 | 1,993 | | 9,012 | 1,652 | | 6,460 | 2,539 | | 8,180 | 4,886 | | 5,060 | 4,812 | |
| | n (%) | n (%) | P–value [a] | n (%) | n (%) | P—value [a] | n (%) | n (%) | P—value [a] | n (%) | n (%) | P—value [a] | n (%) | n (%) | P—value [a] |
| Northeast | 2,440 (21.0) | 471 (17.4) | | 2,229 (20.5) | 449 (21.2) | | 1,514 (17.4) | 708 (18.5) | | 2,273 (22.8) | 1,087 (18.6) | | 1,312 (22.1) | 1,227 (20.1) | |
| Center | 2,641 (38.9) | 578 (39.6) | | 2,554 (38.0) | 437 (36.2) | | 1,903 (40.3) | 756 (43.0) | | 2,119 (36.6) | 1,478 (40.0) | | 1,503 (35.9) | 1,296 (36.4) | |
| South | 2,082 (20.4) | 421 (22.7) | | 2,077 (21.4) | 298 (18.8) | | 1,752 (25.9) | 394 (14.5) | | 1,932 (22.0) | 1,054 (17.9) | | 1,426 (28.6) | 884 (15.1) | |

With VC, the child has a vaccination card and shows it; Without VC, the child does not have a vaccination card, and the child have VC but did not show it; SD, standard Error.

Reported proportions are adjusted to the survey design.

[a] $p$ values were calculated using the Pearson's chi2 test.

## Associated factors

The factors potentially associated with incomplete vaccination schemes are presented in Fig 4. Those who reside in rural areas have 12% (OR 95% CI: 0.7–1.7) more possibility of having an incomplete basic scheme (IBS) in the group of 6 to 11 months compared to the urban area. Also, in households where the caretaker does not have health insurance or has less than elementary schooling, the possibility of having IBS is 40% (OR 95% CI: 1.0–2.0) and 8% (OR 95% CI: 1.0–1.5) higher compared to those with health security or more than secondary education, respectively. Similar patterns for the 12 to 23 months old group were found.

## Discussion

Not showing vaccination card (VC) at survey time has increased significantly in Mexico and other 54 low- and middle-income countries [36]. This situation is relevant because it could bias the coverage estimate and make it difficult the vaccination program monitoring [22, 37, 38]. Some authors maintain that caretakers do not show VCs because they do not identify their usefulness, probably due to health providers who do not request or review them during health care [22]. Others report that mothers without exposure to the media (television, radio, newspapers) and with home birth have a lower likelihood of having and showing VC [37]. Hence, it is important to innovate promotional strategies on the use, conservation and carrying of VC, both for parents and healthcare providers.

According to our study, five main concerns for the Mexican Universal Vaccination Program (UVP) have been found. First, during the analysis period, no estimated coverage exceeded 95%, except for a few vaccines, with the youngest age groups presenting the lowest coverage. That means a longer time at risk of contracting PVD and less protection for the unvaccinated population. Second, many doses of vaccines are applied at the same age; however, the estimated coverage between these doses presents significant differences, which have increased over time and could also be reflecting the critical vaccine shortages [39–42], as well as the rejection of caretakers or healthcare providers to apply multiple doses at the same time. Third, although the rotavirus vaccine is contraindicated in children older than 8 months [4], it is observed that the estimated coverage increases in those older than that age. Fourth, close gaps in vaccination coverage between populations with greater social disadvantage, mainly

**Table 3. Estimated vaccination coverage of two basic schemes according to accumulated doses and age groups, Mexico 2000–2018.**

| | Basic scheme 1 [a] | | | | Basic scheme 2 [b] | | | | | | | |
|---|---|---|---|---|---|---|---|---|---|---|---|---|
| | 2000 | | 2006 | | | 2010 | | 2012 | | 2018 | |
| Vaccine | % | (CI 95%) | % | (CI 95%) | Vaccine | % | (CI 95%) | % | (CI 95%) | % | (CI 95%) |
| **Under 2 months** | | | | | | | | | | | |
| BCG | 96.4 | (94.9–97.9) | 92.9 | (90–95.8) | BCG | 98.1 | (97.3–99) | 92.7 | (90–95.5) | 92.7 | (89.6–95.8) |
| | | | | | 1° dose HB | 83.2 | (77.1–89.3) | 85.3 | (81.8–88.7) | 82.7 | (77.7–87.7) |
| **2–3 months** | | | | | | | | | | | |
| BCG | 97.1 | (96–98.2) | 94.2 | (92.2–96.1) | BCG | 98.5 | (97.9–99.1) | 94 | (92.1–96) | 94 | (91.9–96) |
| 1° dose Sabin | 89.6 | (84.9–94.3) | 65.4 | (58.7–72) | 1° dose HB | 85.9 | (78.7–93) | 87.2 | (80.8–93.6) | 85 | (77.1–92.8) |
| 1° dose PV | 79.1 | (74.5–83.7) | 70.8 | (65.2–76.5) | 2° dose HB | 71.9 | (63.8–80) | 63.3 | (57.7–68.9) | 17.6 | (13.7–21.5) |
| | | | | | 1° dose PVa | 74.1 | (68.3–79.9) | 72.3 | (66.7–77.8) | 65.3 | (59.2–71.4) |
| | | | | | 1° dose Pnm | 62.4 | (51.5–73.2) | 72.3 | (65.5–79) | 89.5 | (86.4–92.6) |
| | | | | | 1° dose RV | 67.2 | (58.6–75.8) | 63.4 | (56.5–70.4) | 81.5 | (76.5–86.6) |
| **4–5 months** | | | | | | | | | | | |
| BCG | 98.2 | (97.4–99) | 96.3 | (94.9–97.8) | BCG | 99.1 | (98.7–99.5) | 96.2 | (94.9–97.6) | 96.2 | (94.7–97.7) |
| 1° dose PV | 91 | (88.1–93.9) | 86.3 | (82.4–90.3) | 1° dose HB | 91.2 | (87.4–94.9) | 91.9 | (88.1–95.6) | 90.5 | (85.6–95.5) |
| 2° dose PV | 89 | (85.8–92.1) | 74.5 | (68.9–80.1) | 2° dose HB | 90.1 | (86–94.3) | 85.7 | (79.7–91.6) | 44 | (32.4–55.5) |
| 1° dose Sabin | 96.6 | (93.3–100) | 85.7 | (78.4–93) | 1° dose Pnm | 82.7 | (77.3–88) | 88.2 | (82.9–93.5) | 96.2 | (94.3–98.2) |
| 2° dose Sabin | 92 | (87.6–96.4) | 71.1 | (64.5–77.6) | 2° dose Pnm | 46.6 | (37.3–56) | 58 | (50.1–65.9) | 77.6 | (70–85.3) |
| | | | | | 1° dose PVa | 90.7 | (84.4–97) | 90.3 | (86.4–94.3) | 87.4 | (81.8–93) |
| | | | | | 2° dose PVa | 68 | (62.1–73.9) | 63.9 | (58–69.9) | 56.9 | (48.9–64.9) |
| | | | | | 1° dose RV | 80.4 | (74.1–86.8) | 77.5 | (68.6–86.4) | 90 | (83.5–96.4) |
| | | | | | 2° dose RV | 57.6 | (48.1–67.1) | 49.9 | (43.3–56.6) | 59.6 | (52.6–66.6) |
| **6–11 months** | | | | | | | | | | | |
| BCG | 98.2 | (97.5–98.9) | 96.4 | (95.3–97.5) | BCG | 99.1 | (98.7–99.4) | 96.3 | (95.3–97.3) | 96.2 | (95.1–97.4) |
| 1° dose PV | 94.7 | (93.5–96) | 91.8 | (89.7–93.9) | 1° dose HB | 89.2 | (85–93.4) | 90 | (87.9–92.2) | 88.5 | (84.5–92.5) |
| 2° dose PV | 93.9 | (92.5–95.3) | 90.4 | (88.3–92.6) | 2° dose HB | 92.5 | (91.1–93.8) | 88.9 | (86.4–91.5) | 53.1 | (46.6–59.6) |
| 3° dose PV | 93 | (91.4–94.5) | 77.8 | (73.3–82.2) | 3° dose HB | 69.7 | (66.6–72.8) | 60 | (54.6–65.4) | 41.5 | (28.9–54.2) |
| 1° dose Sabin | 97.2 | (95.1–99.3) | 87.4 | (82.6–92.2) | 1° dose Pnm | 87.6 | (83.9–91.2) | 91.9 | (89.5–94.2) | 97.5 | (96.5–98.4) |
| 2° dose Sabin | 95.1 | (92.7–97.6) | 90.2 | (86.8–93.7) | 2° dose Pnm | 71.7 | (67.1–76.2) | 80.4 | (73.8–87) | 91.8 | (88.4–95.1) |
| 3° dose Sabin | 92.3 | (90.2–94.5) | 80.5 | (75.6–85.5) | 1° dose PVa | 96.2 | (94.8–97.5) | 95.8 | (93.8–97.8) | 94.2 | (92.2–96.3) |
| | | | | | 2° dose PVa | 87 | (83.5–90.5) | 84.8 | (81.5–88.1) | 81 | (77–85) |
| | | | | | 3° dose PVa | 68 | (60.4–75.5) | 63.8 | (60.7–67) | 61.1 | (57.6–64.6) |
| | | | | | 1° dose RV | 89.1 | (85.5–92.6) | 87.3 | (83.1–91.5) | 94.8 | (92.8–96.7) |
| | | | | | 2° dose RV | 76.2 | (73.6–78.8) | 70.8 | (62.7–78.9) | 77.8 | (74–81.6) |
| **12–23 months** | | | | | | | | | | | |
| BCG | 98.8 | (98.2–99.3) | 97.5 | (96.7–98.2) | BCG | 99.4 | (99.1–99.6) | 97.4 | (96.7–98.1) | 97.4 | (96.7–98.1) |
| MMR | 81 | (71.5–90.5) | 81.3 | (72.1–90.5) | MMR | 67.8 | (53.3–82.3) | 71.5 | (60–83) | 64.9 | (52.5–77.4) |
| 1° dose PV | 98.7 | (98.2–99.2) | 97.9 | (97.2–98.6) | 1° dose HB | 92.6 | (90.8–94.5) | 93.1 | (91–95.1) | 92.1 | (89.6–94.6) |
| 2° dose PV | 98.8 | (98.2–99.3) | 97.8 | (97.1–98.4) | 2° dose HB | 94.9 | (93.3–96.4) | 92.4 | (91.2–93.7) | 61.5 | (58.1–64.9) |
| 3° dose PV | 98.4 | (97.8–98.9) | 93.8 | (91.7–95.8) | 3° dose HB | 87 | (84.7–89.3) | 81.4 | (79.3–83.4) | 67.9 | (58.8–77.1) |
| 1° dose Sabin | 98.8 | (97.5–100) | 93.9 | (90.1–97.8) | 1° dose Pnm | 88.7 | (86–91.5) | 92.7 | (91.2–94.1) | 97.8 | (97.1–98.4) |
| 2° dose Sabin | 98.2 | (96.7–99.7) | 95.5 | (92.8–98.3) | 2° dose Pnm | 79.4 | (76.4–82.3) | 86.5 | (83.7–89.2) | 94.7 | (93.1–96.2) |
| 3° dose Sabin | 97.6 | (96–99.2) | 94.2 | (90.9–97.5) | 1° dose PVa | 97.5 | (96.1–99) | 97.4 | (96.4–98.4) | 96.4 | (95.3–97.5) |
| | | | | | 2° dose PVa | 95.7 | (94.6–96.8) | 95 | (93.7–96.3) | 93.7 | (91.3–96) |
| | | | | | 3° dose PVa | 89.4 | (87.9–91) | 87.6 | (86.1–89.1) | 86.3 | (84.6–87.9) |
| | | | | | 1° dose RV | 89.5 | (86.3–92.8) | 87.8 | (85.5–90) | 95 | (93.6–96.4) |

(*Continued*)

**Table 3.** (Continued)

| Basic scheme 1 [a] | | | | Basic scheme 2 [b] | | | | | |
|---|---|---|---|---|---|---|---|---|---|
| | 2000 | | 2006 | | | 2010 | | 2012 | | 2018 |
| Vaccine | % | (CI 95%) | % | (CI 95%) | Vaccine | % | (CI 95%) | % | (CI 95%) | % | (CI 95%) |
| | | | | | 2° dose RV | 82.4 | (80.3–84.6) | 78.4 | (74.1–82.8) | 83.5 | (81.6–85.4) |

BCG, Bacillus Calmette-Guerin vaccine; PV, complete cells pentavalent vaccine; PVa, acellular pentavalent vaccine; HB, Vaccine against Hepatitis B; Pnm, Conjugate vaccine against Pneumococcal; RV, Vaccine against rotavirus; MMR, Vaccine against measles, mumps, and rubella.

[a] Vaccination scheme valid during the 2000 and 2006 surveys.

[b] Vaccination scheme valid during the 2010, 2012 and 2018. The boost doses were not included (3 Pnm, 3 RV y 4 Pva).

between population with and without health insurance. Fifth, low concordance between the estimated coverage according to the administrative records vs. surveys.

Identification of these concerns can help to propose strategies focused on increasing caretakers´ demand for vaccination [43, 44], quality care by health providers [44, 45], involvement of stakeholders [41, 46, 47], and better program monitoring [5, 6, 9, 48]. Regarding caretakers, it is important to promote the dissemination of key messages about vaccination and to share balanced information about its benefits and harms, which could be presented clearly, simply, and tailored to their needs and culture [21, 43, 44]. On the other hand, specific courses or workshops about indications or contraindications for vaccination [44, 45], and avoiding wasting vaccines and supplies [39, 40, 47] are required, as well as the development of summary clinic guidelines or electronic applications for accompaniment in daily practice [15].

The lowest HB, MMR, and PVa estimated coverages in Mexico could be partially explained by these vaccine shortages, which were reported by PAHO [49] and local authorities [50]. This is relevant given the global context, where problems remain with some vaccines production, limited supplier options [39, 40], and/or recurrent outbreaks, etc. The low and lower-middle-income countries were more affected, and some of them with a serious impact on health care and VPD reemergence [39–42]. In this sense, it is relevant that stakeholders anticipate strategies to purchase vaccines and supplies or at long term, to invest in domestic production.

Strengthening the UVP information monitoring system and linking information systems remains as a relevant recommendation over the past decade [4, 20–22]. Nevertheless, some public and private strategies implemented have presented sustainability important challenges, which may be due to the segmented health system and the consequently weak stewardship, availability of sufficient resources, computing capacity, trained human resources, political will,

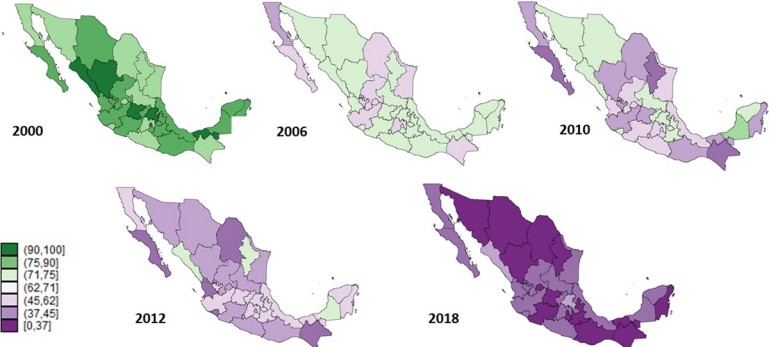

**Fig 2. Estimated coverage of the complete basic scheme in the 6–11 months population, Mexico 2000–2018.**

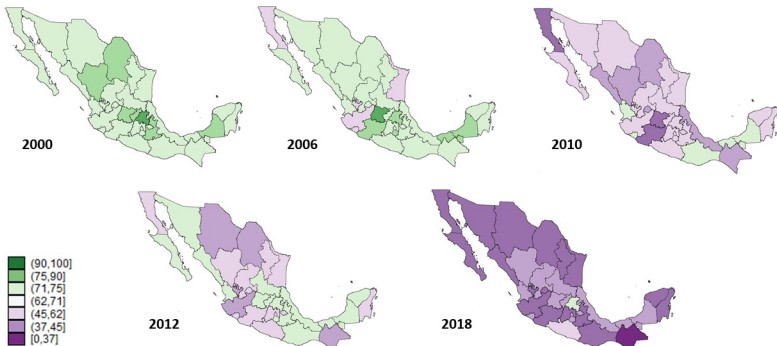

**Fig 3. Estimated coverage of the complete basic scheme in the 12–23 months population, Mexico 2000–2018.**

among others. Hence, to face this concern, it is important to involve key actors and together develop an agenda to get a timely and strong health information system.

The limitations of this study are presented below. First, the way the vaccination schemes information is recorded on the survey forms varies, mainly due to the introduction of new vaccines and the discontinuance of others. Second, the representativeness for the four groups of children under one-year-old may be affected because it was not considered in the survey design. Third, vaccines applied during intensive vaccination weeks were not considered. Fourth, some contextual factors for incomplete schemes (socioeconomic level, vaccine availability, geographic and administrative barriers, etc.) were not included in the study because the surveys did not record that information. Fifth, this study is limited to an analysis with a quantitative approach, to know in depth the reason for this phenomenon; qualitative studies also will be necessary.

In conclusion, the imputation procedure allowed us to recover vaccination information and obtain better estimates, which differed with the previously reported estimation coverage. Additionally, our multi-survey estimates allow coverage to be evaluated according to time, age groups, vaccines, their doses at the national level, and the CBS at the state level. The estimated vaccination coverage increases in older age groups, but decreases for the second or third vaccination doses, except for the pneumococcal and rotavirus vaccine. Furthermore, vaccination coverage does not reach sufficient levels to ensure population immunity. It is necessary to prioritize the population without health insurance, residents in rural areas, and with less schooling.

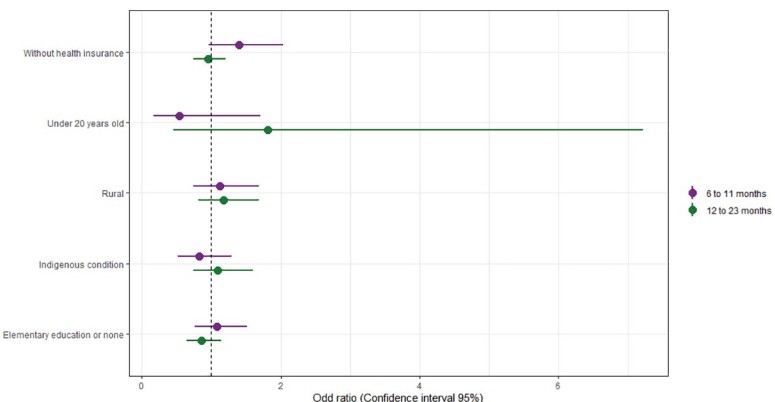

**Fig 4. Factors associated to incomplete basic vaccination schemes by two age groups.**

## Supporting information

**S1 Table. Estimated vaccination coverage of two basic schemes according to accumulated doses and age groups, Mexico 2000–2018.** BCG, Bacillus Calmette-Guerin vaccine; PV, complete cells pentavalent vaccine; PVa, acellular pentavalent vaccine; HB, Vaccine against Hepatitis B; Pnm, Conjugate vaccine against Pneumococcal; RV, Vaccine against rotavirus; MMR, Vaccine against measles, mumps, and rubella. [a] Vaccination scheme valid during the 2000 and 2006 surveys. [b] Vaccination scheme valid during the 2010, 2012 and 2018. The boost doses were not included (3 Pnm, 3 RV y 4 Pva).
(XLSX)

**S2 Table. Estimated vaccination coverage of "with vaccination card" and "without vaccination card" groups by year and age groups, Mexico 2000–2018.** BCG, Bacillus Calmette-Guerin vaccine; PV, complete cells pentavalent vaccine; PVa, acellular pentavalent vaccine; HB, Vaccine against Hepatitis B; Pnm, Conjugate vaccine against Pneumococcal; RV, Vaccine against rotavirus; MMR, Vaccine against measles, mumps, and rubella.
(DOCX)

## Author Contributions

**Conceptualization:** Maria Jesus Rios-Blancas, Hector Lamadrid-Figueroa, Rafael Lozano.

**Data curation:** Maria Jesus Rios-Blancas.

**Formal analysis:** Maria Jesus Rios-Blancas, Hector Lamadrid-Figueroa.

**Investigation:** Maria Jesus Rios-Blancas, Miguel Betancourt-Cravioto, Rafael Lozano.

**Methodology:** Maria Jesus Rios-Blancas, Hector Lamadrid-Figueroa, Rafael Lozano.

**Supervision:** Hector Lamadrid-Figueroa, Miguel Betancourt-Cravioto, Rafael Lozano.

**Validation:** Hector Lamadrid-Figueroa, Miguel Betancourt-Cravioto, Rafael Lozano.

**Visualization:** Rafael Lozano.

**Writing – original draft:** Maria Jesus Rios-Blancas.

**Writing – review & editing:** Hector Lamadrid-Figueroa, Miguel Betancourt-Cravioto, Rafael Lozano.

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
