## [Decision Letter · Decision Letter 0]

29 Jan 2021

PONE-D-20-25678

Vaccination coverage estimation in Mexico in children under five years of age: trends and associated factors

PLOS ONE

Dear Dr. Lamadrid-Figueroa,

Thank you for submitting your manuscript to PLOS ONE. After careful consideration, we feel that it has merit but does not fully meet PLOS ONE’s publication criteria as it currently stands. Therefore, we invite you to submit a revised version of the manuscript that addresses the points raised during the review process.

We look forward to receiving your revised manuscript.

Kind regards,

Holly Seale

Academic Editor

PLOS ONE

Journal Requirements:

2.Please provide additional details regarding participant consent. In the ethics statement in the Methods and online submission information, please ensure that you have specified (1) whether consent was informed and (2) what type you obtained (for instance, written or verbal, and if verbal, how it was documented and witnessed). If your study included minors, state whether you obtained consent from parents or guardians. If the need for consent was waived by the ethics committee, please include this information.

3.Please include your tables as part of your main manuscript and remove the individual files. Please note that supplementary tables (should remain/ be uploaded) as separate "supporting information" files.

4.We note that Figure(s) 2A and 2B in your submission contain map images which may be copyrighted. All PLOS content is published under the Creative Commons Attribution License (CC BY 4.0), which means that the manuscript, images, and Supporting Information files will be freely available online, and any third party is permitted to access, download, copy, distribute, and use these materials in any way, even commercially, with proper attribution. For these reasons, we cannot publish previously copyrighted maps or satellite images created using proprietary data, such as Google software (Google Maps, Street View, and Earth). For more information, see our copyright guidelines: http://journals.plos.org/plosone/s/licenses-and-copyright.

a) You may seek permission from the original copyright holder of Figure(s) 2A and 2B to publish the content specifically under the CC BY 4.0 license. 

Reviewers' comments:

Reviewer's Responses to Questions

**Comments to the Author**

1. Is the manuscript technically sound, and do the data support the conclusions?

Reviewer #1: Yes

2. Has the statistical analysis been performed appropriately and rigorously? 

Reviewer #1: Yes

3. Have the authors made all data underlying the findings in their manuscript fully available?

Reviewer #1: Yes

4. Is the manuscript presented in an intelligible fashion and written in standard English?

Reviewer #1: Yes

5. Review Comments to the Author

Reviewer #1: The manuscript uses MICE methods to estimate the proportion of children vaccinated by age group with each antigen in the routine infant schedule across 5 surveys in Mexico.

Table 1: Normally Hepatitis B after the birth dose is included in pentavalent vaccine. You have it listed separately. Please explain what the 5th antigen in pentavalent is if not hepatitis B?

Table 3: The n and % for sex are identical for ENSA 2000 and 2006. Is this correct?

Introduction 2nd paragraph: Calling EPI "outstanding" and "remarkable." It would be better to offer specific examples to demonstrate/quantify what you are referring to.

Methods: It is unclear if some children are known to be completely unvaccinated and don't have a card for that reason. How were this instances handled in the model?

Results: It would help the reader to know the estimated coverage by age group for each vaccine and survey among children who did present a vaccine card. Additionally, if verbal report from non-card presenting children, this would also be helpful, perhaps in the supplementary materials.

It seems that some birth cohorts would have been considered in more than one survey. Did you consider presenting findings by birth cohort over time?

Overall, there are several places where the language could be simplified or made clearer. One example is sentence 2, paragraph 1 of the introduction.

6. PLOS authors have the option to publish the peer review history of their article (what does this mean?). If published, this will include your full peer review and any attached files.

Reviewer #1: No

---

## [Author Response · Author response to Decision Letter 0]

22 Mar 2021

Response to Journal Requirements and Reviewers

Journal Requirements:

Response: Thank you for the observation. We checked the submitted manuscript and found some PLOS ONE's style mistakes, we apologize for that, we did change in the main manuscript.

2.Please provide additional details regarding participant consent. In the ethics statement in the Methods and online submission information, please ensure that you have specified (1) whether consent was informed and (2) what type you obtained (for instance, written or verbal, and if verbal, how it was documented and witnessed). If your study included minors, state whether you obtained consent from parents or guardians. If the need for consent was waived by the ethics committee, please include this information.

Response: 

We added the following paragraph in the data sources part of the Methodology:

“A secondary analysis from five Mexican national health surveys (2000 [23], 2006 [24], 2010 [25], 2012 [26], and 2018 [27]) was carried out, whose design was probabilistic with national and state representativeness. The surveys content and protocols were approved by the ethics and research committee at the National Institute of Public Health (INSP, by its acronym in Spanish) of Mexico. Data collection was carried out by both INSP and National Institute of Statistic and Geography (INEGI, by its acronym in Spanish), household questionnaires and individual questionnaires were used [23-27]. All participants in the surveys provided oral and/or written consents, where information was provided on the objectives of the research, the voluntary nature of participation and the confidentiality of the information. For children under five years old, mother or guardian (caretaker) were asked to provide information on the vaccination of their children. All the information collected was anonymized and published in digital repositories. More detail was published elsewhere [23-27].”

3.Please include your tables as part of your main manuscript and remove the individual files. Please note that supplementary tables (should remain/ be uploaded) as separate "supporting information" files.

Response: Sincere apologies for the error in the submitted version. We included them in the main revised manuscript.

4.We note that Figure(s) 2A and 2B in your submission contain map images which may be copyrighted. All PLOS content is published under the Creative Commons Attribution License (CC BY 4.0), which means that the manuscript, images, and Supporting Information files will be freely available online, and any third party is permitted to access, download, copy, distribute, and use these materials in any way, even commercially, with proper attribution. For these reasons, we cannot publish previously copyrighted maps or satellite images created using proprietary data, such as Google software (Google Maps, Street View, and Earth). For more information, see our copyright guidelines: http://journals.plos.org/plosone/s/licenses-and-copyright.

a) You may seek permission from the original copyright holder of Figure(s) 2A and 2B to publish the content specifically under the CC BY 4.0 license. 

Response: Thank you for the observation and detailed suggestions. Nevertheless, we elaborated the maps ourselves from the estimates of our study. It should be noted that Figure 2A and 2B are now “Figure 2” and “Figure” 3 in the final manuscript, respectively.

Reviewers' comments:

Reviewer's Responses to Questions

Comments to the Author

1. Is the manuscript technically sound, and do the data support the conclusions?

Reviewer #1: Yes

2. Has the statistical analysis been performed appropriately and rigorously?

Reviewer #1: Yes

3. Have the authors made all data underlying the findings in their manuscript fully available?

Reviewer #1: Yes

4. Is the manuscript presented in an intelligible fashion and written in standard English?

Reviewer #1: Yes

5. Review Comments to the Author

Reviewer #1: 

The manuscript uses MICE methods to estimate the proportion of children vaccinated by age group with each antigen in the routine infant schedule across 5 surveys in Mexico.

Table 1: Normally Hepatitis B after the birth dose is included in pentavalent vaccine. You have it listed separately. Please explain what the 5th antigen in pentavalent is if not hepatitis B?

Response: 

In 1999, The national immunization program has incorporated the pentavalent whole-cell vaccine, which protects against diphtheria, tetanus, pertussis, hepatitis B and Haemophilus influenzae type b. Then, in 2007 the pentavalent whole-cell vaccine was replaced by the acellular pentavalent (PVac), which no longer contains the hepatitis B component and incorporates the inactivated poliomyelitis virus, in such a way that the hepatitis B vaccine is included separately 

Table 3: The n and % for sex are identical for ENSA 2000 and 2006. Is this correct?

Response: Thank you for the observation, it was our mistake, we changed the main manuscript. 

Introduction 2nd paragraph: Calling EPI "outstanding" and "remarkable." It would be better to offer specific examples to demonstrate/quantify what you are referring to.

Response: Thank you for the observation. We changed the main manuscript.

Methods: It is unclear if some children are known to be completely unvaccinated and don't have a card for that reason. How were this instances handled in the model?

Response: 

The caretaker of the children without a vaccination card answered the reminder questions, their self-reported coverage data are presented in S2 Table. It should be noted that the 2000 survey did not record the information of the group that did not have a vaccination card, thus data for this group was not reported in the table.

The surveys do not record the reason why they do not have the vaccination record, however, in a verbal interview with the person in charge of the survey design, he indicated that it was because the mothers left the records in the children's daycare centers. Another reason that could explain this is the increase in social insecurity and therefore the distrust of the population.

We can see that not having a card does not necessarily mean that a child has not been vaccinated, however, the information that is retrieved has many limitations and that when comparing it with the group that did present a card, differences of up to 90 percentage points are identified. In this sense, we decided to handle this group as missing data to estimate its vaccination status through the model. 

Results: It would help the reader to know the estimated coverage by age group for each vaccine and survey among children who did present a vaccine card. Additionally, if verbal report from non-card presenting children, this would also be helpful, perhaps in the supplementary materials. 

Response: 

Thanks for the recommendation, we estimate for both groups, with and without vaccination card. These results can be found at the end of the document, which we also add as supplementary material.

It seems that some birth cohorts would have been considered in more than one survey. Did you consider presenting findings by birth cohort over time?

Response: 

We did not consider it because our population of interest was those under five years of age and the surveys were conducted every six years, except for the 2010 and 2012 surveys.

Overall, there are several places where the language could be simplified or made clearer. One example is sentence 2, paragraph 1 of the introduction.

Response: Thanks for the recommendation, we did an additional review of English.

6. PLOS authors have the option to publish the peer review history of their article (what does this mean?). If published, this will include your full peer review and any attached files.

Do you want your identity to be public for this peer review? For information about this choice, including consent withdrawal, please see our Privacy Policy.

Reviewer #1: No

---

## [Editor Report · Decision Letter 1]

1 Apr 2021

Vaccination coverage estimation in Mexico in children under five years of age: trends and associated factors

PONE-D-20-25678R1

Dear Dr. Lamadrid-Figueroa,

We’re pleased to inform you that your manuscript has been judged scientifically suitable for publication and will be formally accepted for publication once it meets all outstanding technical requirements.

Kind regards,

Holly Seale

Academic Editor

PLOS ONE
---

## [Editor Report · Acceptance letter]

7 Apr 2021

PONE-D-20-25678R1 

Vaccination coverage estimation in Mexico in children under five years old: trends and associated factors 

Dear Dr. Lamadrid-Figueroa:

I'm pleased to inform you that your manuscript has been deemed suitable for publication in PLOS ONE. Congratulations! Your manuscript is now with our production department. 

Kind regards, 

on behalf of

Dr. Holly Seale 

Academic Editor

PLOS ONE